# DISCOVERING CLASSIFICATION RULES FOR INTERPRETABLE LEARNING WITH LINEAR PROGRAMMING

## ABSTRACT

Rules embody a set of if-then statements which include one or more conditions to classify a subset of samples in a dataset. In various applications such classification rules are considered to be interpretable by the decision makers. We introduce two new algorithms for interpretability and learning. Both algorithms take advantage of linear programming, and hence, they are scalable to large data sets. The first algorithm extracts rules for interpretation of trained models that are based on tree/rule ensembles. The second algorithm generates a set of classification rules through a column generation approach. The proposed algorithms return a set of rules along with their optimal weights indicating the importance of each rule for classification. Moreover, our algorithms allow assigning cost coefficients, which could relate to different attributes of the rules, such as; rule lengths, estimator weights, number of false negatives, and so on. Thus, the decision makers can adjust these coefficients to divert the training process and obtain a set of rules that are more appealing for their needs. We have tested the performances of both algorithms on a collection of datasets and presented a case study to elaborate on optimal rule weights. Our results show that a good compromise between interpretability and accuracy can be obtained by the proposed algorithms.

## 1 INTRODUCTION

Medical diagnosis, educational and juridical decisions often have important consequences for the society. Therefore, both the accuracy and interpretability of these decisions are of crucial importance. In particular, these decisions should be understandable by the decision makers. Rule sets consisting of a few intuitively coherent rules have shown to accomplish this purpose in such domains (Lakkaraju et al., 2016). Here, we first aim at rule extraction from powerful ensemble methods and then solely focus on rule generation. In both cases, our objective is to obtain a set of classification rules that balances the trade-off between accuracy and interpretability. Our main tools in this effort are linear programming and column generation.

A rule is an independent if-then statement, which contains one or more conditions that assign a class to a set of samples. For example in binary classification, "**if** (`Clump Thickness` is greater than six) **and** (`Single Epithelial Cell Size` is less than four) **then** the tumor is **malignant**" is such a rule that can be used for breast cancer diagnosis. When a sample satisfies this rule, then it receives a label corresponding to one of the two classes. In case a sample is covered by more than one rule, then majority voting among the assigned labels is used to determine the class of the sample.

Growing decision trees is closely related to rule-based learning. A Decision Tree (DT) naturally results with a set of leaves, where each leaf corresponds to a different rule. On one hand, rule learning is considered to be more flexible for interpretability than tree-based learning approaches. Fürnkranz (1999) lists superiorities of rule-based learning over tree-based learning. Leaves (rules) of a DT are not independent from each other. For example, in a binary tree when a splitting is performed at a node, the left child grows a rule, while its right sibling grows its negation. As a result, every sample obeys to exactly one rule and it is classified according to the corresponding leaf. Thus, inaccurate rules, which are negations of their siblings after splitting, can be created. This

may render false classification of samples and reduce both accuracy and interpretability. However, independently constructed rules do not need such a negation rule, and hence, rule-based learning can be considered more flexible for interpretability. On the other hand, unlike DTs, an independent set of rules does not necessarily cover the entire sample space. This may result in a state where a test sample can not be classified with the proposed rule set. This drawback is often handled by assigning a default label to those uncovered samples (Fürnkranz et al., 2012). To minimize the number of uncovered samples during testing, the training data is required to be fully covered. Separate-and-conquer algorithm by Fürnkranz (1999) achieves this heuristically by fitting rules on uncovered training samples, and new rules are generated until each training sample is covered by at least one rule. Instead of such a sequential covering, we take advantage of our linear programming (LP) approach and explicitly impose a covering constraint on the training set.

In this paper, we propose two algorithms for interpretability and learning that are based on mathematical programming. We give a generic LP model that is first used by our Rule Extraction (RUX) algorithm, which selects rules from trained tree or rule ensembles for interpretation. Then, we develop a Rule Generation (RUG) algorithm, which generates classification rules using column generation (CG) to solve the LP model.

Rule extraction methods attempt to select a set of rules from accurate complex or black-box models to interpret the predictions of these models (Hayashi & Oishi, 2018). For instance, several works in the literature aim at interpreting Random Forest (RF) models by extracting rules from the trees in the forest (Liu et al., 2012; Lu Thi et al., 2015; Adnan & Islam, 2017; Wang et al., 2020; Birbil et al., 2020). Most of these studies use heuristic approaches to select the desired set of rules. Birbil et al. (2020) suggest a set covering formulation to extract intrepretable rules from RFs. Other applications include extraction of rules from artificial neural networks (Andrews et al., 1995) and support vector machines (Barakat & Bradley, 2010). As a remark, there are other post-hoc interpretability approaches in the literature such as SHAP (Lundberg & Lee, 2017) and LIME (Ribeiro et al., 2016) that are considered to be model-agnostic. However, our RUX algorithm is a model-specific approach.

There are several studies that are closely related to ours, since they also employ mathematical programming for rule learning. Malioutov & Varshney (2013) propose a rule-based binary classifier solving a binary program that minimizes the number of rules for a boolean compressed sensing problem. Wang & Rudin (2015) present a mixed integer linear programming (MILP) formulation to learn decision rules for binary classification (*e.g.*, patterns). They give a discussion on how their classifier is equivalent to DTs and RFs. The MILP formulation is solved using generated rules with an objective of minimizing the number of misclassified samples, the number of rules generated, and the total length of each rule. Dash et al. (2020) offer a CG-based framework to find optimal rule set for binary classification, where the objective is to find a trade-off between classification rule simplicity and accuracy. One-hot encoding is used to binarize categorical data, and numerical data is also discretized with sample deciles as thresholds. For large instances, the pricing subproblem is either solved with time limits, or the model columns are generated by a greedy heuristic. Wei et al. (2019) propose generalized linear rule models using a similar CG framework of the work by Dash et al. (2020). Malioutov & Meel (2018) solve a Max-Sat formulation by constraint programming to construct interpretable classification rules. Ghosh & Meel (2019) also propose a framework based on MaxSAT formulation that can be applied to binary classification problems with binary features to minimize the number of generated rules and the number of misclassified samples. Their approach is incremental and takes advantage of partitioning the data set into several clusters.

**Contributions.** RUX and RUG are based on an LP model and thus, both algorithms are scalable for large datasets. This is an important advantage compared to the existing studies that use MILP formulations. The proposed algorithms directly address multi-class problems while the existing studies using optimization-based approaches are for binary classification. To that end, the objective function in our LP formulation minimizes classification error using a loss function instead of explicitly counting misclassified samples. Both RUX and RUG can work with continuous or categorical features, and hence, they do not require encoding of the data. Along with the set of rules, our algorithms also return the optimal weights for the rules. These weights allow to attach importance to each rule for interpreting the classification. Our algorithms admit assigning cost coefficients to the rules. These coefficients could relate to different attributes of the rules, such as; rule lengths, estimator weights, number of false negatives, and so on. The objective function also allows penaliz-

ing rules that may have undesired outcomes (like long rule lengths). Thus, the decision makers can use these coefficients to lead the training process to obtain a set of rules that are more appealing for their needs. The novelty in our column generation approach is the use of a regular decision tree with sample weights as the pricing subproblem. Training trees with sample weights is very fast, and also standard in all machine learning packages as boosting methods also rely on sample weights. Lastly, we present our algorithms for multi-class classification problems. However, the proposed ideas can also be extended to discovering regression rules for interpretable learning with linear programming.

## 2 RULE EXTRACTION

We consider a classification problem with $K$ classes and denote the set of class labels by $\mathcal{K}$. The training dataset consists of samples with features $\boldsymbol{x}_i \in \mathbb{R}^p$ for $i \in \mathcal{I}$ and labels $y_i$ for $i \in \mathcal{I}$. To work with multiple classes, we define a vector-valued mapping $\boldsymbol{y}(\boldsymbol{x}_i) \in \mathcal{K} \subset \mathbb{R}^K$ as in Zhu et al. (2009). That is, if $y_i = k$, then

$$\boldsymbol{y}(\boldsymbol{x}_i) = (-\tfrac{1}{K-1}, -\tfrac{1}{K-1}, \dots, 1, \dots, -\tfrac{1}{K-1})^\intercal, \tag{1}$$

where the value one appears only at the $k^{th}$ component of the vector.

Suppose that we have a collection of rules indexed by $\mathcal{J}$. A rule $j \in \mathcal{J}$ assigns the vector $\boldsymbol{R}_j(\boldsymbol{x}_i) \in \mathcal{K}$ to input $\boldsymbol{x}_i$, only if rule $j$ covers sample $i$. This vector is also formed in the same manner as in equation 1. To predict the class of a given sample $\boldsymbol{x}_i$ with this collection of rules, we use a set of nonnegative weights $w_j, j \in \mathcal{J}$ associated with the rules and evaluate

$$\hat{\boldsymbol{y}}(\boldsymbol{x}_i) = \sum_{j \in \mathcal{J}} a_{ij} \boldsymbol{R}_j(\boldsymbol{x}_i) w_j, \tag{2}$$

where $a_{ij} \in \{0, 1\}$ indicates whether rule $j$ covers sample $i$ or not. Then, the index of the largest component of the resulting vector $\hat{\boldsymbol{y}}(\boldsymbol{x}_i)$ is assigned as the predicted label $\hat{y}_i$ of sample $i \in \mathcal{I}$. Note that equation 2 is similar to the weighting of the classifiers in standard boosting methods. Here, instead of classifiers, we use rules for classifying only the *covered* samples.

In order to evaluate the classification error, we use the *hinge loss* and define the total classification loss by

$$\sum_{i \in \mathcal{I}} \max \big\{ 1 - \sum_{j \in \mathcal{J}} \hat{a}_{ij} w_j, 0 \big\},$$

where $\hat{a}_{ij} = \kappa a_{ij} \boldsymbol{R}_j(\boldsymbol{x}_i)^\intercal \boldsymbol{y}(\boldsymbol{x}_i)$ with $\kappa = (K-1)/K$. This loss function allows us to write a linear programming model, where the objective is to find the set of rules that minimizes the total loss. To this end, we introduce the auxiliary variables $v_i, i \in \mathcal{I}$ standing for $v_i \geq \max \big\{ 1 - \sum_{j \in \mathcal{J}} \hat{a}_{ij} w_j, 0 \big\}$, and obtain our *master problem*

$$
\begin{aligned}
\text{minimize} \quad & \sum_{i \in \mathcal{I}} v_i + \sum_{j \in \mathcal{J}} c_j w_j \\
\text{subject to} \quad & \sum_{j \in \mathcal{J}} \hat{a}_{ij} w_j + v_i \geq 1, \quad i \in \mathcal{I}; \\
& \sum_{j \in \mathcal{J}} a_{ij} w_j \geq \varepsilon, \qquad i \in \mathcal{I}; \\
& v_i \geq 0, \qquad\qquad\quad i \in \mathcal{I}; \\
& w_j \geq 0, \qquad\qquad\quad j \in \mathcal{J},
\end{aligned}
\tag{3}
$$

where $c_j \geq 0$, $j \in \mathcal{J}$ are the cost coefficients. These coefficients and the second set of constraints require further explanation. The cost coefficients serve two important roles: First, solutions involving many rules with nonzero weights are avoided. The less the number of rules in the resulting set, the easier it is to interpret a solution. In other words, we prefer sparse solutions and $c_j$ serves to that preference. Second, in many different application domains, rules have actual costs that need to be taken into consideration. As we also highlight in our title, when interpretability is of concern, one could try to obtain rules with few features because shorter rules are considered to be easier to interpret (Lakkaraju et al., 2016). In this case, the number of conditions in a rule can be set as the rule cost. As another example, consider a classification problem in medical diagnosis, where false negatives are likely to be perceived as more costly than false positives. Such an evaluation could also be easily incorporated with a rule cost in the proposed model. We should point out that there

exists a trade-off between model accuracy and the rule set size. This can also be handled with our formulation by introducing a fixed cost coefficient, *i.e.*, using the same value for all $c_j$, $j \in \mathcal{J}$.

The master problem equation 3 also involves a set of *covering constraints* with the fixed right-hand-side, $\varepsilon > 0$. These constraints make sure that each sample is covered by at least one rule. The need for these covering constraints is exemplified as follows:

> *Consider a binary classification problem, where we select three samples $\{\boldsymbol{x}_i, \boldsymbol{x}_k, \boldsymbol{x}_l\}$ along with their labels $y_i = 1$, $y_k = -1$ and $y_l = -1$. Suppose that we have two rules $j$ and $j'$ such that the former covers all three, whereas the latter covers only the last two samples. Here, the rules use majority voting as it is applied to the leaves in trained DTs. The labels assigned by the rules $j$ and $j'$ are as follows: $R_j(\boldsymbol{x}_i) = R_j(\boldsymbol{x}_k) = R_j(\boldsymbol{x}_l) = -1$ and $R_{j'}(\boldsymbol{x}_k) = R_{j'}(\boldsymbol{x}_l) = -1$. The first set of constraints in equation 3 then becomes*

$$
\begin{array}{lllll}
-a_{ij}w_j & & +v_i & & \geq 1, \\
a_{kj}w_j & +a_{kj'}w_{j'} & & +v_k & \geq 1, \\
a_{lj}w_j & +a_{lj'}w_{j'} & & & +v_l \geq 1.
\end{array}
$$

> *For simplicity, if we also assume that $c_j = c_{j'}$, then the optimal solution becomes $w_j = v_k = v_l = 0$ and $w_{j'} = v_i = 1$. With this solution, sample $\boldsymbol{x}_i$ is not covered.*

We remark that covering each sample in a dataset is crucial from the perspective of giving a literal interpretation with the obtained rules. This is particularly important when the resulting set of rules is used to interpret the classification of individual samples in the dataset. Clearly, varying $\varepsilon$ may lead to a change in the set of optimal rules. For instance a large $\varepsilon$ value may impose larger rule weights on those samples covered with only few rules. However, we point out that the role of this constraint is just coverage, and hence, setting $\varepsilon$ to a small strictly positive value is sufficient[1].

Up until this point, we have not specified any details about the rule set $\mathcal{J}$. On one hand, this rule set can be *static* in the sense that it can be obtained by extracting the rules available through a trained tree ensemble algorithm or by using the rules resulting from a rule-based method (Cohen & Singer, 1999). For instance, consider a RF model trained on a given dataset. Then, the rule set $\mathcal{J}$ can be obtained from the leaves of the trees in the forest, since each leaf corresponds to a rule. Solving equation 3 with such a rule set allows us to extract the rules that were most critical for classification. As we have mentioned before, we can assign the length of a rule as its cost and try to obtain rules with desirable lengths for interpretation. In a similar vein, consider another example with a trained AdaBoost (Freund & Schapire, 1997) model for which the base estimators are set as DTs. Again, the leaves of the trees from AdaBoost can be used to construct the rule set $\mathcal{J}$ in equation 3, which is then solved to extract an interpretable set of rules. The costs of the rules in this case could be the inverse of the estimator weights assigned by the AdaBoost algorithm to the trees. In this way, the obtained set of rules is more likely to inherit the importance of the DTs from the AdaBoost model. Using these trained models to construct our master problem leads to our first algorithm RUX, which is based on tree ensembles. We show in Section 4 that RUX can indeed extract only a selection of rules from the trained models without significantly sacrificing accuracy. The computational complexity for RUX is determined by the underlying LP solver used. In practice, most of the LP solvers use interior point methods that work in polynomial-time.

## 3    RULE GENERATION

Suppose now that we do not have the entire set of rules $\mathcal{J}$ explicitly, or the size of this collection is too large to be constructed by an existing learning framework. Notice that, the universe of rules $\mathcal{J}$ contains exponentially many rules due to all possible combinations of different levels of the considered features. This aspect shows the combinatorial structure of the problem. Thus, the rules should be generated in an iterative fashion. Since equation 3 is a linear programming problem

---

[1]In case one insists on treating $\varepsilon$ as a hyperparameter and proceeds with tuning, it is important to know that there will be intervals of $\varepsilon$ values where the optimal set of rules remains the same. We have given this parametric analysis in Section A.2 of the supplementary document. Using this analysis, the tuning can be conducted in a computationally efficient manner.

and rules correspond to columns, this iterative scheme leads to the well-known column generation approach in optimization (Desaulniers et al., 2006). At each iteration of column generation, a linear programming model is constructed with a subset of the columns (*column pool*) of the overall model. This model is called the *restricted master problem*. After solving the restricted master problem, the dual optimal solution is obtained. Then using this dual solution, a *pricing subproblem* is solved to identify the columns with negative *reduced costs*. These columns are the only candidates for improving the objective function value when they are added to the column pool. The next iteration continues after extending the column pool with the negative reduced cost columns.

In order to apply column generation to our problem, we define a subset of rules $\mathcal{J}_t \subset \mathcal{J}$ at iteration $t$ and form the restricted master problem by replacing $\mathcal{J}$ with $\mathcal{J}_t$ in equation 3. Let us denote the dual variables associated with the first set of constraints by $\beta_i$, $i \in \mathcal{I}$. Likewise, let $\gamma_i$, $i \in \mathcal{I}$ be the dual variables corresponding to the coverage constraints. In vector notation, we simply use $\boldsymbol{\beta}$ and $\boldsymbol{\gamma}$. Then, the *dual restricted master problem* at iteration $t$ becomes

$$
\begin{aligned}
\text{maximize} \quad & \sum_{i \in \mathcal{I}} (\beta_i + \varepsilon \gamma_i) \\
\text{subject to} \quad & \sum_{i \in \mathcal{I}} (\hat{a}_{ij} \beta_i + a_{ij} \gamma_i) \leq c_j, \quad j \in \mathcal{J}_t; \\
& 0 \leq \beta_i \leq 1, \quad i \in \mathcal{I}; \\
& \gamma_i \geq 0, \quad i \in \mathcal{I}.
\end{aligned}
\tag{4}
$$

If we denote the optimal dual solution at iteration $t$ by $\boldsymbol{\beta}^{(t)}$ and $\boldsymbol{\gamma}^{(t)}$, then improving the objective function value of problem equation 4 requires finding at least one rule $j' \in \mathcal{J}/\mathcal{J}_t$ such that

$$
\bar{c}_{j'} = c_{j'} - \sum_{i \in \mathcal{I}} \left( \hat{a}_{ij'} \beta_i^{(t)} + a_{ij'} \gamma_i^{(t)} \right) < 0,
\tag{5}
$$

where $\bar{c}_{j'}$ is the reduced cost of column $j'$. In fact, this condition simply checks whether $j' \in \mathcal{J}/\mathcal{J}_t$ violates the dual feasibility. To find those rules with negative reduced costs, we formulate the pricing pricing subproblem as

$$
\max_{j \in \mathcal{J}/\mathcal{J}_t} \left\{ \sum_{i \in \mathcal{I}} \hat{a}_{ij} \beta_i^{(t)} + a_{ij} \gamma_i^{(t)} \right\}.
\tag{6}
$$

Linear programming theory ensures that if the pricing subproblem does not return any rule with a negative reduced cost, then we have the optimal solution to our master problem equation 3 with the current set of rules, $\mathcal{J}_t$. Otherwise, we have at least one rule with a negative reduced cost. After adding one or more rules with negative reduced costs to $\mathcal{J}_t$, we proceed with $\mathcal{J}_{t+1}$ and solve the restricted master problem or, equivalently, its dual equation 4. Algorithm 3 shows the steps of the exact rule generation approach. It is important to note that the sole purpose of solving pricing subproblem equation 6 is to return a subset of rules with negative reduced costs. We denote this subset by $\mathcal{J}_- \subseteq \mathcal{J}/\mathcal{J}_t$.

[!h] Exact Rule Generation **Input:** training data, $(\boldsymbol{x}_i, y_i)_{i \in \mathcal{I}}$   $t = 0$   Construct initial rule pool, $\mathcal{J}_0$ True $(\boldsymbol{\beta}^{(t)}, \boldsymbol{\gamma}^{(t)}) \leftarrow$ Solve equation 4   $\mathcal{J}_- \leftarrow$ Solve pricing subproblem, equation 6   $\mathcal{J}_- = \emptyset$ **return** $\mathcal{J}_t$ $t \leftarrow t + 1$   $\mathcal{J}_t = \mathcal{J}_{t-1} \cup \mathcal{J}_-$

We have overlooked two important steps in Algorithm 3. The first one is the construction of the initial rule pool $\mathcal{J}_0$ (line three). One possible solution for this step is to train a DT and use its leaves as the starting set of rules. The second step is solving the pricing subproblem (line six). Note that solving equation 6 is associated with finding a rule with the highest accuracy when the samples have *weights* $\beta_i^{(t)}$, $i \in \mathcal{I}$. As constructing an optimal binary decision tree is known to be $\mathcal{NP}$-complete (Laurent & Rivest, 1976), solving problem equation 6 is rather difficult even when $\boldsymbol{\gamma}^{(t)} = \boldsymbol{0}$. This leads us to the proxy pricing subproblem of column generation as shown in Figure 1. Here, treating $\boldsymbol{\gamma_i} = \boldsymbol{0}$ allows the construction of a DT using the dual values $\boldsymbol{\beta}^{(t)}$ as sample weights. DTs can be solved very quickly using standard libraries that are available in all machine learning packages. Therefore, we propose to train a DT as a heuristic pricing approach, where the sample weights correspond to dual variables $\beta_i^{(t)}$, $i \in \mathcal{I}$. Then, we can check whether any leaf of the DT satisfies equation 5 before adding those rules to the current rule pool. Notice that a negative reduced cost column (rule) with $\boldsymbol{\gamma_i} = \boldsymbol{0}$ would also have a negative reduced cost when the pricing subproblem is solved exactly.

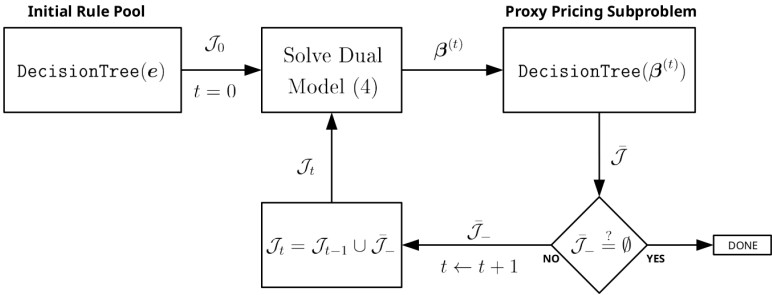

Figure 1: Proposed rule generation algorithm.

Figure 1 shows the proposed heuristic approach for rule generation. The procedure DecisionTree takes a vector of sample weights as an input and returns a set of rules $\bar{\mathcal{J}}$ (leaves of the decision tree). The vector $e$ is the vector of ones. The subset $\bar{\mathcal{J}}_- \subseteq \bar{\mathcal{J}}$ contains those rules with negative reduced costs satisfying equation 5. As we use the dual solution $\gamma^{(t)}$ only for checking the reduced cost, training decision trees with sample weights in this manner boils down to solving a *proxy pricing subproblem*. Indeed, our rule generation algorithm RUG does not guarantee to solve the overall problem equation 3 to optimality. Fortunately, this may be to our advantage since a sub-optimal solution can rectify the problem of overfitting to the training data. In fact, overfitting due to exact optimality is a common concern also in recent optimization-based approaches to learning (Günlük et al., 2018; Verwer & Zhang, 2019). Avid readers of optimization-based learning would also notice that there are some connections between RUG and the boosting methods. We elaborate on this point in Section A.3 of the supplementary document.

We remark once again that growing decision trees with sample weights is a standard tool in every library on machine learning. Likewise, there are various options to solve linear programs. Both growing DTs and solving LPs can be achieved in polynomial-time. Therefore, the proposed algorithm in Figure 3 is extremely easy to implement with the existing packages of popular programming languages.

## 4 COMPUTATIONAL STUDY

We have evaluated the performances of the proposed algorithms on a collection of frequently used datasets. The details of these datasets are given in Section A.1 of the supplementary document. Both the rule extraction (RUX) and the rule generation (RUG) algorithms are implemented in Python [2]. Since our computation times do not exceed one minute even on a standard laptop computer, we do not report them here. We have used stratified $10 \times 3$ nested cross validation to estimate the generalization performance of the tested methods. In all experiments, the right-hand-side value of each covering constraint is set to $\varepsilon = 0.01$ .

### 4.1 NUMERICAL EXPERIMENTS

First, we apply our RUX algorithm to trained random forest (RF) and AdaBoost (ADA) models. For both RF and ADA, we have selected the maximum-depth parameter from the set $\{1, 2, 3\}$ and the number-of-estimators parameter from the set $\{100, 200\}$. After obtaining the best parameter set, we have used the corresponding trained model to apply RUX. We denote the models after applying RUX to RF and ADA by RUX-RF and RUX-ADA, respectively. In RUX-RF, we have used the rule-length as the cost coefficients $c_j$, $j \in \mathcal{J}$ in equation 3. For RUX-ADA, the cost coefficients are assigned as the inverse of the estimator weights of the trees in the trained ADA model. That is, the rules coming from the same tree receive the same cost coefficient.

---

[2]Our implementation is available in the supplementary file.

Table 1 summarizes our results. Overall, we observe that RUX-RF and RUX-ADA give accuracies on par with RF and ADA. In fact, RUX-RF can even outperform RF on several datasets. These are BANKNOTE, TICTACTOE, ADULT, BANK-MKT, MAGIC, MUSHROOM, MUSK, PHONEME and MAMMOGRAPHY. On average, RUX-RF and RUX-ADA obtain far fewer number of rules than their counterparts RF and ADA, respectively. When we compare RUX variants, we observe that RUX-ADA generally obtains smaller sets with shorter rule lengths than RUX-RF. This is due to the larger trees preferred by RF models as the average number of rules obtained with RF is higher than ADA. Take for instance GLASS and SENSORLESS, for which ADA returns more rules than RF. For those instances, we clearly see that the average rule length resulting from RUX-RF is smaller than those from RUX-ADA.

Table 1: The performances of Random Forest (RF), AdaBoost (ADA) as well as Rule Extraction applied to RF and ADA - denoted as RUX-RF and RUX-ADA. In the table, *ACC.*, *# RULES*, and *RULE LEN.* stand for accuracy, number of rules and rule length, respectively. The first value in each cell gives the mean, whereas the value in parenthesis gives the standard deviation.

| DATASET | RF | | ADA | | RUX-RF | | | RUX-ADA | | |
| --- | --- | --- | --- | --- | --- | --- | --- | --- | --- | --- |
| | ACC. | # RULES | ACC. | # RULES | ACC. | # RULES | RULE LEN. | ACC. | #RULES | RULE LEN. |
| BANKNOTE | 0.955 (0.018) | 1009.8 (379.538) | **0.999** (0.003) | 667.8 (358.674) | 0.991 (0.008) | 24.2 (6.844) | 2.118 (0.081) | 0.995 (0.006) | 36.6 (15.493) | 1.864 (0.681) |
| HEARTS | **0.832** (0.059) | 751.1 (496.415) | 0.815 (0.069) | 220.0 (63.246) | 0.809 (0.080) | 55.3 (32.575) | 2.094 (0.824) | 0.799 (0.080) | 28.9 (3.9) | 1.000 (0.000) |
| ILPD | **0.713** (0.014) | 664.0 (526.334) | 0.700 (0.059) | 655.8 (322.577) | 0.696 (0.048) | 69.6 (58.064) | 1.778 (0.742) | 0.703 (0.062) | 90.2 (49.198) | 2.020 (0.644) |
| IONOSPHERE | 0.929 (0.041) | 997.3 (322.732) | **0.934** (0.041) | 779.3 (436.083) | 0.912 (0.051) | 32.7 (9.9) | 2.277 (0.162) | 0.906 (0.036) | 67.9 (17.748) | 2.317 (0.744) |
| LIVER | 0.695 (0.060) | 962.0 (349.497) | **0.747** (0.075) | 330.3 (172.687) | 0.692 (0.083) | 110.6 (11.138) | 2.375 (0.201) | 0.698 (0.091) | 37.5 (20.823) | 1.155 (0.327) |
| PIMA | 0.759 (0.033) | 1265.7 (409.449) | **0.760** (0.034) | 415.7 (144.178) | 0.742 (0.048) | 173.5 (25.418) | 2.643 (0.055) | 0.753 (0.037) | 55.6 (22.481) | 1.390 (0.412) |
| TICTACTOE | 0.768 (0.049) | 1280.0 (413.118) | **1.000** (0.000) | 1360.0 (386.437) | 0.998 (0.004) | 100.0 (12.824) | 3.000 (0.000) | **1.000** (0.000) | 79.0 (6.074) | 3.000 (0.000) |
| TRANSFUSION | 0.766 (0.025) | 745.6 (122.06) | **0.794** (0.030) | 404.8 (156.886) | 0.769 (0.031) | 34.5 (14.864) | 2.158 (0.217) | 0.766 (0.018) | 12.3 (6.667) | 1.263 (0.465) |
| WDBC | 0.953 (0.030) | 1060.3 (385.359) | **0.970** (0.026) | 612.0 (390.43) | 0.954 (0.017) | 50.7 (3.773) | 2.632 (0.244) | 0.963 (0.027) | 50.6 (9.812) | 1.689 (0.806) |
| ADULT | 0.798 (0.004) | 1587.6 (4.088) | **0.866** (0.004) | 1450.0 (27.125) | 0.857 (0.004) | 111.1 (53.532) | 2.850 (0.055) | 0.860 (0.003) | 72.7 (7.973) | 2.495 (0.026) |
| BANK_MKT | 0.781 (0.014) | 1507.1 (250.918) | 0.844 (0.012) | 1271.2 (195.308) | 0.843 (0.012) | 337.6 (30.376) | 2.854 (0.019) | **0.854** (0.007) | 235.2 (47.098) | 2.600 (0.267) |
| MAGIC | 0.804 (0.007) | 1278.1 (412.559) | 0.872 (0.004) | 1437.8 (48.26) | 0.850 (0.011) | 244.7 (39.26) | 2.740 (0.019) | **0.875** (0.008) | 310.8 (21.181) | 2.658 (0.017) |
| MUSHROOM | 0.981 (0.008) | 1033.4 (328.702) | **1.000** (0.000) | 441.3 (115.858) | **1.000** (0.000) | 11.6 (4.006) | 2.478 (0.165) | **1.000** (0.000) | 12.2 (2.44) | 1.683 (0.794) |
| MUSK | 0.904 (0.007) | 1262.4 (407.446) | **0.994** (0.003) | 1513.6 (8.168) | 0.970 (0.007) | 183.6 (23.491) | 2.949 (0.015) | 0.978 (0.007) | 206.7 (22.186) | 2.950 (0.010) |
| OILSPILL | 0.957 (0.000) | 1093.1 (465.142) | **0.970** (0.011) | 1098.5 (447.0) | 0.963 (0.018) | 60.7 (19.574) | 2.519 (0.536) | 0.964 (0.014) | 61.0 (6.65) | 2.763 (0.277) |
| PHONEME | 0.800 (0.017) | 1353.8 (385.843) | 0.858 (0.030) | 1202.7 (46.55) | 0.845 (0.024) | 168.3 (28.81) | 2.565 (0.032) | **0.865** (0.018) | 188.4 (10.091) | 2.463 (0.035) |
| MAMMOGRAPHY | 0.983 (0.002) | 1259.0 (406.919) | **0.987** (0.003) | 1026.3 (413.774) | 0.986 (0.002) | 153.7 (21.401) | 2.540 (0.044) | 0.986 (0.003) | 108.2 (23.266) | 2.290 (0.207) |
| SEEDS | 0.910 (0.094) | 803.5 (400.555) | **0.933** (0.078) | 739.4 (435.589) | 0.876 (0.108) | 17.3 (7.056) | 1.783 (0.356) | 0.886 (0.159) | 17.1 (8.399) | 2.207 (0.592) |
| WINE | **0.978** (0.039) | 685.9 (360.995) | 0.955 (0.051) | 558.6 (226.358) | 0.938 (0.049) | 19.4 (2.221) | 1.905 (0.235) | 0.939 (0.041) | 22.3 (6.55) | 1.954 (0.419) |
| GLASS | 0.687 (0.072) | 1262.7 (355.563) | **0.794** (0.066) | 1324.1 (373.676) | 0.679 (0.078) | 45.8 (5.116) | 2.364 (0.101) | 0.716 (0.113) | 55.6 (6.753) | 2.742 (0.048) |
| ECOLI | **0.854** (0.055) | 1164.2 (411.335) | 0.836 (0.050) | 917.2 (445.408) | 0.795 (0.080) | 41.4 (6.168) | 2.087 (0.105) | 0.812 (0.035) | 42.1 (7.593) | 2.600 (0.343) |
| SENSORLESS | 0.768 (0.008) | 1456.2 (12.985) | **0.892** (0.014) | 1562.6 (10.752) | 0.541 (0.041) | 78.2 (5.633) | 2.559 (0.053) | 0.632 (0.071) | 101.4 (6.45) | 2.818 (0.034) |
| AVERAGE | 0.844 (0.030) | 1112.855 (345.798) | 0.887 (0.030) | 908.591 (237.501) | 0.850 (0.037) | 96.568 (19.184) | 2.421 (0.194) | 0.861 (0.038) | 86.014 (14.947) | 2.178 (0.325) |

We report our results with the proposed RUG algorithm in Table 2. In addition to RF and ADA, we have also added a DT algorithm for comparison. The same set of values $\{1, 2, 3\}$ is used for tuning the `maximum-depth` parameter both in DT and RUG (for constructing the initial column pool and the proxy pricing subproblem; see Figure 1). In RUG, we have used the rule length as cost coefficients. When we compare the accuracies of different methods, we observe that the average performance of RUG is better than those of DT and RF, and close to ADA. Since the resulting models from DT are interpretable, we also report the rule numbers as well as the rule lengths for DT. RUG outperforms DT in terms of accuracy at the expense of generating more rules. However, the average lengths of the rules obtained with RUG are shorter than those obtained with DT in all instances.

In their recent work, Dash et al. (2020) propose a rule learning approach based on column generation. They also give a comparison against other studies and present that their algorithm shows one of the best performances. We compare their results against a variant of RUG in Table 4 of the supplementary document (Section A.4). We remark that their numerical study is limited to binary classification instances, and hence, the benchmarking is conducted only on a subset of datasets that we have used in our previous two tables. On average, our results are on par with the results obtained with their method in terms of accuracy and interpretability. It is important to note that both RUX variants and RUG are an order of magnitude faster on large datasets than the approach of Dash et al. (2020). Section A.4 of the supplementary document involves a further discussion.

When it comes to the scalability of our algorithms, our preliminary results on large instances are very promising and in favor of RUG in terms of both accuracy and interpretability. RUX variants also surpass the accuracy of their corresponding trained models from both RF and ADA. We also observe that RUG outputs comparable computational times with RF and ADA. Our results are presented and discussed in detail in Section A.6 of the supplementary document.

Even though RUX variants and RUG obtain few number of rules with short lengths for many datasets, there are also several datasets where the resulting sets of rules are still challenging to interpret. However, note that the results that we present in Table 1 and Table 2 are given for *all* rules with *nonzero* optimal weights. Using a threshold weight for a rule and monitoring the accuracy,

Table 2: The performances of Random Forest (RF), AdaBoost (ADA), Decision Tree (DT) and Rule Generation (RUG). Here, *ACC.*, *# RULES*, and *RULE LEN.* stand for accuracy, number of rules, and rule length, respectively. The first value in each cell gives the mean, whereas the value in parenthesis gives the standard deviation.

| DATASET | RF | ADA | DT | | | RUG | | |
|---|---|---|---|---|---|---|---|---|
| | ACC. | ACC. | ACC. | #RULES | RULE LEN. | ACC. | #RULES | RULE LEN. |
| BANKNOTE | 0.955 (0.018) | **0.999** (0.003) | 0.937 (0.017) | 8.0 (0.0) | 3.0 (0.0) | **0.999** (0.003) | 42.9 (6.871) | 2.063 (0.236) |
| HEARTS | **0.832** (0.059) | 0.815 (0.069) | 0.775 (0.041) | 7.4 (1.897) | 2.8 (0.632) | 0.802 (0.066) | 18.7 (6.038) | 1.169 (0.358) |
| ILPD | 0.713 (0.014) | 0.700 (0.059) | **0.715** (0.009) | 2.0 (0.0) | 1.0 (0.0) | 0.689 (0.042) | 34.4 (27.253) | 1.492 (0.665) |
| IONOSPHERE | 0.929 (0.041) | **0.934** (0.041) | 0.897 (0.051) | 5.1 (1.729) | 2.5 (0.527) | 0.912 (0.039) | 41.6 (10.997) | 1.802 (0.624) |
| LIVER | 0.695 (0.060) | **0.747** (0.075) | 0.675 (0.068) | 5.4 (2.319) | 2.3 (0.675) | 0.680 (0.089) | 33.8 (26.41) | 1.289 (0.481) |
| PIMA | 0.759 (0.033) | **0.760** (0.034) | 0.738 (0.037) | 4.4 (1.265) | 2.1 (0.316) | 0.747 (0.057) | 26.8 (13.206) | 1.241 (0.39) |
| TICTACTOE | 0.768 (0.049) | **1.000** (0.000) | 0.714 (0.034) | 8.0 (0.0) | 3.0 (0.0) | 0.968 (0.031) | 50.5 (10.865) | 3.0 (0.0) |
| TRANSFUSION | 0.766 (0.025) | **0.794** (0.030) | 0.758 (0.035) | 6.0 (2.667) | 2.4 (0.843) | 0.775 (0.033) | 35.1 (23.187) | 1.771 (0.454) |
| WDBC | 0.953 (0.030) | **0.970** (0.026) | 0.919 (0.030) | 7.0 (1.633) | 2.8 (0.422) | 0.965 (0.025) | 39.6 (11.711) | 2.034 (0.452) |
| ADULT | 0.798 (0.004) | **0.866** (0.004) | 0.844 (0.005) | 8.0 (0.0) | 3.0 (0.0) | 0.859 (0.004) | 51.6 (24.213) | 2.321 (0.153) |
| BANK_MKT | 0.781 (0.014) | 0.844 (0.012) | 0.775 (0.008) | 8.0 (0.0) | 3.0 (0.0) | **0.852** (0.010) | 114.5 (13.142) | 2.629 (0.058) |
| MAGIC | 0.804 (0.007) | **0.872** (0.004) | 0.787 (0.009) | 7.2 (1.687) | 2.8 (0.422) | 0.865 (0.007) | 96.4 (23.172) | 2.47 (0.303) |
| MUSHROOM | 0.981 (0.008) | **1.000** (0.000) | 0.985 (0.004) | 7.0 (0.0) | 3.0 (0.0) | **1.000** (0.001) | 11.2 (4.59) | 2.138 (0.377) |
| MUSK | 0.904 (0.007) | **0.994** (0.003) | 0.915 (0.011) | 8.0 (0.0) | 3.0 (0.0) | 0.940 (0.007) | 86.4 (7.058) | 2.78 (0.081) |
| OILSPILL | 0.957 (0.000) | **0.970** (0.011) | 0.963 (0.013) | 4.8 (1.687) | 2.2 (0.422) | 0.962 (0.010) | 36.3 (9.557) | 1.642 (0.507) |
| PHONEME | 0.800 (0.017) | 0.858 (0.034) | 0.769 (0.020) | 6.0 (2.108) | 2.5 (0.527) | **0.863** (0.021) | 93.1 (13.503) | 2.318 (0.078) |
| MAMMOGRAPHY | 0.983 (0.002) | **0.987** (0.003) | 0.984 (0.002) | 6.8 (1.932) | 2.7 (0.483) | 0.986 (0.003) | 62.3 (20.271) | 2.318 (0.28) |
| SEEDS | 0.910 (0.094) | **0.933** (0.078) | 0.890 (0.081) | 6.2 (1.317) | 2.8 (0.422) | 0.914 (0.100) | 17.2 (10.13) | 2.029 (0.209) |
| WINE | **0.978** (0.039) | 0.955 (0.051) | 0.904 (0.071) | 7.6 (0.966) | 3.0 (0.0) | 0.950 (0.067) | 19.0 (6.342) | 1.52 (0.314) |
| GLASS | 0.687 (0.072) | **0.794** (0.066) | 0.687 (0.056) | 7.7 (0.675) | 3.0 (0.0) | 0.640 (0.084) | 29.3 (10.457) | 2.545 (0.207) |
| ECOLI | **0.854** (0.055) | 0.836 (0.050) | 0.804 (0.041) | 8.0 (0.0) | 3.0 (0.0) | 0.816 (0.062) | 16.5 (5.061) | 2.498 (0.211) |
| SENSORLESS | 0.768 (0.008) | **0.892** (0.014) | 0.425 (0.002) | 6.0 (0.0) | 3.0 (0.0) | 0.690 (0.038) | 27.6 (4.427) | 2.744 (0.097) |
| AVERAGE | 0.844 (0.030) | 0.887 (0.030) | 0.812 (0.029) | 6.573 (0.995) | 2.677 (0.259) | 0.858 (0.036) | 44.764 (13.112) | 2.082 (0.297) |

one can end up with a set of rules that is easier to interpret. We elaborate on this point in the next subsection with a case study.

## 4.2 CASE STUDY

We demonstrate an interpretation of RUG results using "Breast Cancer Wisconsin (Original)" dataset (Dua & Graff, 2017). This dataset has nine features with integer values from $[1, 10]$ and two diagnosis classes (benign or malignant). RUG reaches the accuracy level of 0.96 and generates 18 rules that we denote as $R_1$ to $R_{18}$. Figure 2 shows that RUG can also attain accuracy level of 0.95 using only the first 12 rules. The plot given in Figure 2 be used by decision makers (DMs) to interpret the rules and their weights obtained with RUG. The horizontal axis lists the rules in descending order of their optimal weights assigned by RUG. Here, the rule weights shown on the left side of the vertical axis are normalized by dividing the weights with the largest among them. On the right side, we present the accuracy of RUG on the *test data* cumulatively adding one rule at a time in the model. In other words, using only the first rule results with the accuracy of 0.92. With the addition of second and third rules, the accuracy level increases to 0.93. This can be traced following the blue line and the corresponding point for each rule. When rule twelve is added, the accuracy of RUG catches that of the RF. As a remark, there is a slight deterioration in the accuracy when rule four is included and it rebounds back with addition of rule eleven. This can be expected since the performance is evaluated on the test data. Above each bar in the plot, we give the cumulative percentage of test instances covered by the rules. That is, rule one covers 73% of the test data and coverage gradually increases with the addition of subsequent rules. The coverage of samples reaches to 95% with only three rules and when the first nine rules are used, the coverage becomes full. Overall, interpretability plot is a playground for DMs to choose the best trade-off between accuracy and interpretability depending on the sensitivity of the application. Further details on our case study is presented in Section A.5 of the supplementary document.

## 5 CONCLUSION

We have proposed a linear programming (LP) approach to discover a set of critical rules in multiclass classification. The objective of this LP is to find a set of rule weights such that the sum of the classification error and the total rule cost is minimized. Using the optimal weights of the LP, a set of interpretable rules is obtained that can be used for classification. This approach has led to two algorithms that are both simple, fast and easy-to-implement. The numerical experiments have shown that the proposed algorithms may be used for hitting the right balance between accuracy and interpretability.

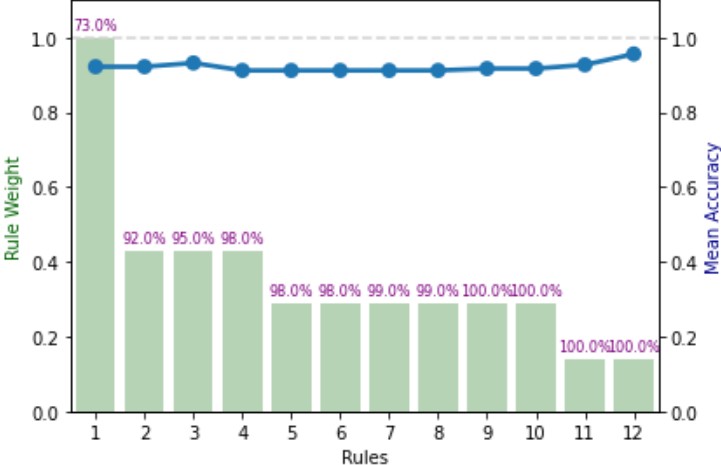

Figure 2: Interpretation plot. RUG surpasses accuracy level of DT by using the first 3 rules that ordered in terms of their normalized weights (bar heights). The percentages show the *cumulative* fractions of the samples covered after adding each rule.

The first algorithm works with a predefined set of rules. To demonstrate our algorithm, we have used the models trained by Random Forest and AdaBoost algorithms. These two algorithms are frequently used for their remarkable accuracy. However, it is very difficult to interpret the trained models as both algorithms return tree ensembles involving thousands of rules. With Random Forest, we have used the rule lengths as cost coefficients in our algorithm to obtain shorter rules for interpretation. As AdaBoost returns also a set of estimator weights, we have used these weights as cost coefficients to mimic the performance of the trained AdaBoost model. Our numerical results show that the performance of our rule extraction algorithm is close to –for several datasets even better than– its ensemble counterparts. More importantly, this performance comes with much fewer rules than the rules produced with both Random Forest and AdaBoost. On one hand, the set of rules required to apply our rule extraction algorithm may result from tree ensembles. On the other hand, this set may also come from other learning methods such as rule ensembles that produce a large collection of rules. Testing our algorithm further with those rule ensembles is an interesting future research direction.

Instead of a given collection of rules, our second algorithm uses a rule generation mechanism that iteratively solves LPs and builds up a set of rules. This algorithm has a solid mathematical basis as its core idea comes from the well-known column generation approach in large-scale linear programming. A significant advantage of this algorithm is having more control on the generation of the rules. However, there is a challenge that lies within the difficult exact pricing subproblem. As a remedy, we have proposed to solve a proxy subproblem by training decision trees with sample weights. These sample weights are obtained from the dual of the LPs at that iteration. Decision trees can be easily trained with the existing software packages, and hence, this proxy pricing problem can be implemented quite efficiently. We have compared the performance of our rule generation algorithm with the highly interpretable Decision Tree algorithm, and with the highly accurate tree ensemble algorithms, Random Forest and AdaBoost. At the expense of more but shorter rules than Decision Trees, our rule generation algorithm performs on par with tree ensemble algorithms in terms of accuracy.

This study is conducted for multi-class classification. With a piecewise-linear loss function (*e.g.* mean absolute deviation), our LP model can also be reformulated for regression problems. An alternative approach can be to incorporate the dual values corresponding to coverage constraints within the solution of the proxy pricing subproblem. This might require developing a specially tailored DT that use all the dual values. These are other lines of research that we plan to pursue in the future.

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
