# OpenReview forum: "Discovering Classification Rules for Interpretable Learning with Linear Programming"
_ICLR.cc/2022/Conference — ICLR 2022 Submitted_

### Official Review · Reviewer_fGw9 · 2021-11-02

**Correctness:** 2
**Technical Novelty And Significance:** 2
**Empirical Novelty And Significance:** 1
**Recommendation:** 3
**Confidence:** 3

**Main Review:**

From a conceptual point-of-view, it is unclear what is gained by using linear programming in the proposed manner: runtime is not shown to be improved compared to classic greedy rule learning algorithms and optimality is not preserved in either new algorithm variant because heuristic tree learning algorithms are used to provide the starting point for rule simplification and are also used within the algorithm in the second variant of the proposed approach.

The empirical results do not show an advantage compared to classic rule learning. The main text of the paper only compares to decision tree learning, which is not the appropriate comparison. There are results for some datasets for other rule learners in an appendix, which are obtained from other publications, but these were almost certainly not obtained using exactly the same experimental methodology. (Notably, assuming the methodology is in fact the same, the proposed algorithm is actually outperformed by the classic rule learning algorithm RIPPER: accuracy of RIPPER is slightly better on average, and the rule sets are generally much smaller; also RIPPER does not require a voting mechanism to obtain classifications, which simplifies the interpretation.)

Almost all the datasets in the experiments are two-class problems, and the results indicate that the proposed method performs poorly on the five multi-class problems included in the experiments compared to AdaBoost (which is a boosting algorithm not particularly suited to multi-class problems either). The comparison to other rule learners in the appendix is restricted to two-class problems. This means there is little evidence to show that the proposed algorithm performs well on multi-class problems.

There is a comparison to decision tree learning in Table 3, which the second variant of the proposed approach outperforms, but the decision tree is crippled by setting its maximum depth to three!

Similarly, the comparison to random forests is problematic because trees of a maximum depth of three are used in them. This is fine in the case of boosting (because boosting works well with weak learners), but it is not appropriate for random forests.

Given that the paper employs AdaBoost, it should compare to SLIPPER from "A Simple, Fast, and Effective Rule Learner", which is based on the confidence-rated version of the boosting algorithm.

Another rule learner that should be compared against is FURIA, which was shown to perform better than RIPPER:

Hühn, J., Hüllermeier, E. FURIA: an algorithm for unordered fuzzy rule induction. Data Min Knowl Disc 19, 293–319 (2009)

There is no excuse for not running experiments under exactly the same experimental conditions because code is available and experiments can be run quickly. The authors should also consider using repeated 10-fold cross-validation because most of the datasets are quite small. A single 10-fold cross-validation run produces an estimate of high variance for such datasets.

The case study is based on the very old breast cancer dataset from the UCI repository. Why not use a modern dataset that is of greater interest to people?

Other comments:

* Table 1: "RUX-RF and RUX-ADA give accuracies on par with RF and ADA" - no, for many datasets, accuracy is quite noticeably reduced: ionosphere, liver, transfusion, wdbc, musk, oilspill, seeds, wine, glass, sensorless(!).

* Table 2: "average performance is close to ... ADA" - no, for many datasets, accuracy is quite noticeably reduced and the average accuracy is substantially lower.

* "solving LPs can be achieved in polynomial time" - please state the time complexity of the full rule learning algorithm. How does it compare to that of an algorithm based on classic separate-and-conquer rule learning?

* "one of the two classes" - not necessarily

* "then, majority voting ... is used" - not necessarily

* "the training data is required to be fully covered" - not necessarily, the majority class can be covered by a default rule, for example; or a hybrid nearest-neighbour-based approach can be employed

* "the best parameter set" - this is done using the internal 3-fold cross-validation? Please spell this out.

**Summary Of The Paper:**

The paper presents an approach to learning ensembles of classification rules using linear programming. Two variants of the algorithm are considered: one uses a collection of rules extracted from ensembles of decision trees as the starting point, and another one that learns rules from scratch by applying decision tree learning within the rule learning algorithm. UCI datasets (mostly two-class problems) are used to compare classifier complexity and accuracy with that of random forests, AdaBoost, and learning a single decision tree.

**Summary Of The Review:**

The paper does not show convincingly that the proposed methods improve on existing rule learning algorithms.

---

### Official Review · Reviewer_Pz5i · 2021-11-02

**Correctness:** 3
**Technical Novelty And Significance:** 2
**Empirical Novelty And Significance:** 2
**Recommendation:** 6
**Confidence:** 3

**Main Review:**

Strength: an easy to implement method to select some rules from an existing set of rules, or by iteratively generating them through decision trees. Using LP allows for some scalability, even if modern MILP and SAT solvers could handle quite large instance (and certainly the thousands of rules considered in the experiments).

Weaknesses: the method does not really "learn" a rule system, but rather select a subset of rules from existing rules (either batch-given or generated on the fly). It also starts from the premises that a rule system with as few short rules as possible will ensure interpretability, which is something I would disagree on. This may be considered as necessary, but I would not say sufficient.

Questions:

* The authors compare their methods to other reduction techniques that start from a large set of rules, however there is a big body of work that concentrates on producing directly compact set of rules. How does the proposed method compare to such approaches? For instance, authors could compare to the work "A Bayesian Framework for Learning Rule Sets for Interpretable Classification" that they do cite in the paper. One could also think of older works focusing on rule systems and interpretability, such as fuzzy ones (e.g.,"Guillaume, Serge. "Designing fuzzy inference systems from data: An interpretability-oriented review." IEEE Transactions on fuzzy systems 9.3 (2001): 426-443.")

* Author claim that scalability is an essential feature of the considered approach, yet most considered problems count at most around 1000 rules, a scale that can be easily handled by modern MILP and SAT solvers. The question is then to know if in practice, when one starts from existing rules to select, scalability is such an important aspect?

* While conciseness and compactness is certainly desirable for interpretability, and henceforth to explainability, I think interpretability goes beyond that. For instance, majority voting is mentioned by the authors when multiple rules having different conclusions apply to an example. How in practice do we ensure interpretability in those cases where contradictory information is present? Is there a limit to the number of rules that can be activated by an example?

* The parameter $c_i$, which is a scalar, is supposed to be able to include a lot, and a lot of aspects such as rule length, rule complexity, rule accuracy, etc. Could it be explained how, in practice, one could include all these aspects in one number? In the experiment, only rule length is considered, and I am curious about how $c_i$ should be assessed when other aspects should be included in it?

* A recent trend to produce explanations from trees, random forest and other classifiers through the use of logical formulas (e.g., prime implicants), with the idea of producing logically readable explanation as to why an example is classified the way it is. See for example "Audemard, Gilles, et al. "Trading Complexity for Sparsity in Random Forest Explanations." arXiv preprint arXiv:2108.05276 (2021)." for a recent paper going along this trend. How is the current method positioned with respect to such works?

Other comments:

* The paper could have benefited from a final read/check. For instance, P5 algorithm 1 does not compile well (probably due to the floating environment).

* The paper is not always fully accurate in its statements. For instance, it is inaccurate to say that a sibling growth the negation of a rule, it only includes the negation of the induced (Boolean) literal. Strictly speaking, one rule is not the negation of the other. Similarly, features are sometimes mentioned as belonging to the real-valued hyper-cube (start of section 2), sometimes to be categorical (start of section 3).

**Summary Of The Paper:**

The paper discusses how, from an existing set of rules or a procedure generating rules "on the fly", one can use linear programming to select a subset of rules that would increase "interpretability", where an interpretable system of rules is here understood as a system with few, short rules.

The idea is to ensure a good accuracy level by minimizing a continuous hinge loss (ensuring the linearity of the problem), while penalizing retained rule (rules having a positive weight) through a cost proportional to the complexity of those rules. The experiments show that the paper does, indeed, improve upon the metric retained by the authors (rule length and rule quantity). Whether this produces something that is readable is not checked further on (assuming that this is sufficient for user-readability).

**Summary Of The Review:**

The proposed method does accomplish its intent, which is to reduce a rule system or incrementally integrate rules so that the final system is rather concise and compact. It does not, however, learn the rule system itself, mainly selecting rules that are learned by an external method (in this sense, the title may be misleading). One thing that is not clear is whether the interpretability goal is achieved (but then most papers on interpretability do not go till checking actual interpretability), and how the method would compare to approaches that directly aim at building a compact, "interpretable" rule base?

The paper could also benefit from some small corrections, but is overall quite understandable.

---

### Official Review · Reviewer_a6cV · 2021-11-07

**Correctness:** 3
**Technical Novelty And Significance:** 2
**Empirical Novelty And Significance:** 2
**Recommendation:** 3
**Confidence:** 2

**Main Review:**


Even though the idea in the paper is quite interesting and the paper is well-written, I find the experimental section insufficient to show the method's superiority in comparison to related algorithms.

1. RandomForest to obtain good results often require component tress with max_depth higher than 3. Therefore, claims that the method obtains similar results as RF are - for the moment - not fully supported since the hyperparametrizatoin of RF is very much suboptimal.
2. The proposed method for extracting decision rules from ensembles only selects them and has no mechanism of simplifying them. Despite the fact that the method allows for weighting rules according to their length, in practice, ensembles are often a collection of large decision trees with long decision rules. It can be hard to construct a good classifier with short rules from a large tree without any simplifying mechanism. Currently presented experiments do not tackle this issue.
3. In the experiments with AdaBoost, it would be interesting to see the method's results without providing the information about rule weight calculated by boosting (through cost coefficient). The proposed method calculates the weights automatically, the weights from boosting are no measure of interpretability - therefore, the proposed method should be able to calculate the correct rule weights on its own.
4. In my opinion, the differences in Accuracy between AdaBoost and extracted rules are quite considerable. Even though on average it is -2,6%, there is a drop of 25% on SENSORLESS and several drops of 8-5%. What are the possible reasons of poor algorithm performance on some datasets? Moreover, the method for extraction rules is not compared against any other method of this type.
4. The comparisons in the main paper should include some modern methods for constructing small, interpretable decision rule sets rather than comparisons with standard baselines. Unfortunately, the comparison with other SOTA method in the Appendix does not clearly demonstrate the superiority of the proposed method. For instance, CG on average constructs significantly less complex and slightly more accurate rules. The method's advantage is that it also works on multi-class data; however, the authors should show some comparisons on multi-class data that would prove its usefulness in this setup (besides comparing it to standard, not-interpretable ensembles)

Minor issues:
- The related works could include Angelino et al. CORELS algorithm that constructs optimal decision rules.
- The formatting of Algorithm 3 is incorrect (some Latex error, I guess)


**Summary Of The Paper:**

The paper presents a simple method for constructing decision rules for classification problems through linear programming. In fact, the method rather uses rules generated by the standard DecisionTree algorithm but later selects the most useful rules, weights instances, and poses a new learning problem for the Decision Tree algorithm.


**Summary Of The Review:**

The idea presented in the paper is simple but quite interesting. Unfortunately, I find the experimental section insufficient to show the method's superiority in comparison to related algorithms.

---

### Comment · Area_Chair_3eTo · 2021-11-10
**Discussion phase**

Dear Authors and Reviewers,

Let me first thank you for supporting ICLR 2022: Authors for submitting their contributions, and Reviewers for going through them and sending their comments and remarks!

As the discussion phase has just begun, let me ask Authors to answer all questions appearing in reviews and to defend your paper, and reviewers to check all other reviews to see whether you coincide and to be ready to respond to authors rebuttals.

We are also looking forward for public comments. I hope for a vivid discussion for this paper.

Best regards, AC for Paper 2980

---

### Decision · Program_Chairs · 2022-01-20

**Decision:**

Reject

**Comment:**

The reviewers are rather critical about the paper and the authors did not take a part in the discussion phase. Let me also add that the paper ignores a vast number of papers dealing with a similar problem. The column generation algorithm is a core of LPBooting also used for rule learning ("Rule Learning with Monotonicity Constraints", "The Linear Programming Set Covering Machine"). There are many other papers also using linear relaxation of integer programming to build rule models. Logical analysis of Data is also a well-known method being close to such approaches. There is also a plenty other rule learning systems that should be compared in the experimental study such as Ripper, Slipper, MLRules, or Ender (to mention only a few of them).